# Epidemiology of *Echinococcus granulosus sensu lato* in the Greater Horn of Africa: A systematic review

Weldegebrial G. Aregawi[1,2,3]*, Bruno Levecke[2], Hagos Ashenafi[4], Charles Byaruhanga[5], Nigatu Kebede[4], Erastus Mulinge[6], Marion Wassermann[7], Thomas Romig[7], Pierre Dorny[3], Veronique Dermauw[3]

1 Ethiopian Institute of Agricultural Research, Addis Ababa, Ethiopia, 2 Department of Translational Physiology, Infectiology and Public Health, Faculty of Veterinary Medicine, Ghent University, Ghent, Belgium, 3 Department of Biomedical Sciences, Institute of Tropical Medicine, Antwerp, Belgium, 4 Unit of Animal Health and Zoonotic Diseases Research, Addis Ababa University, Aklilu Lemma Institute of Pathobiology, Addis Ababa, Ethiopia, 5 Department of Veterinary Tropical Diseases, Faculty of Veterinary Science, University of Pretoria, Pretoria, South Africa, 6 Kenya Medical Research Institute, Nairobi, Kenya, 7 Parasitology Unit, University of Hohenheim, Stuttgart, Germany

* weldedr77@gmail.com

**Editor:** jong-Yil Chai, Seoul National University College of Medicine, REPUBLIC OF KOREA

**Data Availability Statement:** All relevant data are within the paper and its Supporting information files. Systematic review registration: The protocol for this review was registered on the International

## Abstract

### Background

Cystic echinococcosis (CE) is a neglected zoonotic disease that is caused by *Echinococcus granulosus sensu lato (s.l.)*, the life cycle of which involves multiple hosts. We conducted a systematic review (SR) on *E. granulosus s.l.* in the Greater Horn of Africa (GHA), to provide a picture of its recent epidemiology across all hosts.

### Methods

For this SR, conducted in accordance with the Preferred Reporting Items for Systematic Reviews and Meta-Analyses (PRISMA) statement, five electronic databases, as well experts in the region were consulted to retrieve records published between 2000 and 2022, reporting the presence of *E. granulosus s.l.* infections in any natural host in the GHA (Djibouti, Eritrea, Ethiopia, Kenya, Sudan, Somalia, South Sudan, Tanzania and Uganda).

### Principal findings

A total of 247 records were retained, describing the presence of *E. granulosus s.l.* throughout the GHA, except for Djibouti. Only few population surveys on human CE were conducted in the area, with the prevalence ranging between 0.3 and 11.3%. In animals, the reported prevalence ranged up to 61.6% in camels, 88.4% in cattle; 65.2% in goats, 9.9% in pigs, 67.8% in sheep and 94.5% in dogs. In addition, *E. granulosus s.l.* was also reported in wildlife. A total of five species were reported in the different hosts, namely *E. granulosus sensu stricto* (G1, G3, G_{Omo}), *E. canadensis* (G6/7), *E. ortleppi* (G5), *E. felidis*, and *E. equinus* (G4).

Prospective Register of Systematic Reviews, PROSPERO: CRD42021288237 (Available from: https://www.crd.york.ac.uk/prospero/display_record.php?ID=CRD42021288237).

**Funding:** This study is supported by the Special Research Fund – Ghent University (https://www.ugent.be/en/research/funding/bof/doc) through a PhD scholarship awarded to WGA with grant number BOF21/DOS/056. The funders had no role in study design, data collection and analysis, decision to publish, or preparation of the manuscript.

**Competing interests:** The authors declare that they have no competing interests.

## Conclusions

We confirm that *E. granulosus s.l.* is prevalent throughout the GHA. Nevertheless, despite our efforts to screen grey literature, an accurate assessment of the epidemiology in GHA remains challenging, due to the lack of combined host, in-depth risk factor and behavioural studies, as well as the wide diversity in subpopulations studied and diagnostic tools used. Interdisciplinary and transboundary partnerships would be essential for the design of effective control strategies, tuned to the GHA setting.

## Author summary

Cystic echinococcosis (CE) is a disease caused by the tapeworm *Echinococcus granulosus* that affects both humans and animals. In the Greater Horn of Africa (GHA), the livelihoods of millions of people depend on livestock and herding dogs. At the same time, poor animal slaughter practices still exist in the area. Therefore, the GHA is considered a textbook example for the transmission of *E. granulosus*. Nevertheless, although the World Health Organization (WHO) prioritized CE as one of the seven neglected zoonotic tropical diseases, a recent summary on the true extent to which the tapeworm occurs across the different host species in the GHA, and the factors that contribute to its transmission is missing. Our systematic review of the literature on the topic confirms that the tapeworm is widely spread in the GHA. At the same time, it remains difficult to accurately assess the occurrence of the worm and how exactly it spreads between the hosts, in absence of well-designed surveys. Based on the nature of the disease (involving both animals and humans) and the disease being highly prevalent in pastoralist communities that move across national borders, a firm response to control CE will require both interdisciplinary and transboundary partnerships.

## Background

Cystic echinococcosis (CE) is a neglected zoonotic disease that is caused by the tapeworm species complex of *Echinococcus granulosus sensu lato (s.l.)*. Five species, some with several genotypes are currently recognized to belong to this complex: *E. granulosus sensu stricto* (genotypes (G) G1, G3, $G_{Omo}$ with sheep and buffalos as main intermediate hosts), *E. equinus* (G4 with horse as main intermediate host), *E. ortleppi* (G5 with cattle as main intermediate host), *E. canadensis* (G6/7 and G8/10 with camels and pigs, and cervids, respectively as main intermediate hosts) and *E. felidis* (lion strain with lions as the definitive host and wild herbivores and omnivores as intermediate hosts) [1,2]. *E. granulosus s.s.* is the major zoonotic species worldwide. It accounts for 88.5% of all human CE cases, followed by *E. canadensis* (11.0%). Few human infections caused by *E. ortleppi* have also been documented worldwide [3].

The natural life cycle of *E. granulosus s.l.* involves carnivores (canids and also certain felids) as definitive hosts, while livestock (e.g., sheep, goat, buffalo, horse, cattle, pig and camel) and wild herbivores/omnivores (e.g., giraffe, kudu, warthogs, zebra) act as intermediate hosts. Through access to infected offal or predation, the definitive hosts ingest hydatid cysts containing protoscolices, which leads to the development of adult worms in their intestines. The intermediate hosts acquire the infection upon ingestion of eggs passed in the feces of infected definitive hosts [4]. Humans can act as dead-end intermediate hosts following accidental

ingestion of tapeworm eggs, causing CE. Globally, it is estimated that the prevalence of human CE cases in endemic areas ranges from 1 to 7% [5], and that 188,000 new cases are reported each year [6]. Since 2020, CE has become one of the seven zoonotic neglected tropical diseases that are prioritized by the WHO, with the ultimate goal to control the disease burden by 2030 [7]. Next to the public health impact, infections in livestock result in major economic losses due to condemnation of offal, and reduction in carcass weight, milk production, fecundity and skin value, leading to an estimated 2 billion US$ production losses globally every year [8].

The Greater Horn of Africa (GHA; Djibouti, Eritrea, Ethiopia, Kenya, Sudan, Somalia, South Sudan, Tanzania and Uganda) [9] is an area that seems highly afflicted by human CE, at least based on previous summaries of mostly outdated studies [10]. In the GHA, the livelihood of millions of people is exclusively dependent on livestock [11]. Moreover, poor animal slaughter practices prevail in the area (e.g., slaughtering takes place in community environments and dogs are fed with hydatid cyst-infected organs) [12–15]. This combined with the large number of herding dogs, the seasonal migration of nomadic pastoral communities with their large herds, and the movement of wild animals (prey and predators) over the shared borders in the area, support the fact that the GHA could be considered a textbook example for the transmission of *E. granulosus s.l.*

A sound understanding of the epidemiology of *E. granulosus s.l.* infections in all hosts is a key factor in reducing disease transmission across the different hosts. Moreover, considering the similar disease ecosystems, livestock trade routes and management systems, it is paramount to investigate the disease epidemiology among the different countries in the GHA. Up to now, existing literature reviews lack an in-depth summary of recent knowledge on the epidemiology of *E. granulosus s.l.* infections in all the hosts combined [16]. The present study was therefore designed to describe the recent epidemiological situation of *E. granulosus s.l.* infections across all hosts in the GHA following a systematic review (SR), with the ultimate aim to improve the knowledge about *E. granulosus s.l.* prevalence in all hosts, the geographical distribution, species/genotypes circulating in the region, and the risk factors associated with its occurrence in the region.

## Methods

### Study design

The SR was conducted in accordance with the Preferred Reporting Items for Systematic Reviews and Meta-Analyses (PRISMA) statement [17] (S1 Checklist). A bibliographic search as well as a search of local and unpublished sources was performed, aiming to collect information on the epidemiology of *E. granulosus s.l.* infections across all hosts in the GHA published from January 2000 to October 2022. The GHA was defined as the region encompassing Djibouti, Eritrea, Ethiopia, Kenya, Somalia, Sudan, South Sudan, Tanzania, and Uganda, which combined covers six million km$^2$. The region is home to more than 350 million people [18], representing ~25% of Africa's total population.

### Search strategy

The bibliographic search was performed on five databases (ScienceDirect, PubMed, CAB Abstracts, Web of Science and Scopus) using the search phrase (Echinococc* OR Hydatid* OR "*E. granulosus*") AND (Djibouti OR Eritrea OR Ethiopia OR Kenya OR Somalia OR "South Sudan" OR Sudan OR Tanzania OR Uganda OR "Horn of Africa"). The online search was initially conducted on November 30, 2021 and again updated on October 18, 2022 in order to identify studies that had been published since the initial search. To identify additional references, pre-identified experts in the subject area were consulted and requested to provide

relevant resources such as, local journals, conference proceedings and university databases. This information was collected using a preformatted data collection sheet (S2 Checklist), and subsequently screened for additional relevant data. Moreover, Google Scholar was searched using the search phrase: "cystic echinococcosis" AND hydatidosis AND "*Echinococcus granulosus*" for each of the included countries, this entailed screening the first few pages for additional records. Finally, reference lists of relevant review articles were also screened for additional records.

### Eligibility criteria

Records were included in the final dataset provided that they met all the following inclusion criteria: the study presented data on (i) any diagnostic test result for *E. granulosus s.l.* in any natural host species; (ii) occurrence, incidence, prevalence and geographical distribution of *E. granulosus s.l.*; (iii) it reported findings for the GHA (Ethiopia, Somalia, Eritrea, Djibouti, Kenya, Uganda, Sudan, South Sudan, and Tanzania); and (iv) it was published between January 1, 2000 and October 18, 2022. Reports describing human cases diagnosed outside GHA, yet providing clear indications that the case originated from the GHA, were included. Similarly, population surveys for livestock examined in a non-GHA country, but that were exported from a GHA country, were included. Records were excluded if they did not meet the above-mentioned criteria or if they met the additional exclusion criteria: (i) the language was not English or French, (ii) the topic was outside the scope of the review (e.g., laboratory experiment or review article), (iii) no full text was available, and (iv) it was a duplicate record.

### Screening process and data extraction

Databases with retrieved records from the different sources were merged and screened for duplicates. Subsequently, titles and abstracts were screened to identify relevant records. Next, the full texts of the retained studies were retrieved and screened for eligibility. This process was run by two independent reviewers (WA and VD). Inconsistent results were discussed until consensus was reached. From the included studies, data were extracted by one member of the review team (WA) using a pre-designed Excel data extraction sheet. The data extraction was subsequently checked by a second member of the team (VD). Inconsistent results were discussed until consensus was reached.

For all included records, bibliographic information such as, author's name, reference, and publication year were extracted. Next, detailed data were extracted for all the studies described in the records. It is important to note that a single bibliographic record could provide data on multiple host species, countries, diagnostic tests, study designs or study period. As each combination provides unique data, we refer to a study to make this explicit. In other words, one record may have multiple studies. For each study, host species, study country, diagnostic test(s), study design and study period were extracted. In addition, data on the species/genotypes, the affected organs and fertility of cysts were extracted when available. Regarding the country, studies performed in Sudan published before 2011 were by default categorised under Sudan, as South Sudan became an independent country in that year. For population surveys, we additionally extracted the study area, the sample size, the number of positive cases and the calculated prevalence. For population surveys reporting on putative risk factors, the number of exposed and non-exposed individuals, in the diseased and non-diseased groups were also noted, for each of the risk factors reported as being significantly associated with disease. For case reports, we extracted sex, age, affected organ, as well as information about signs and/or symptoms, applied treatment and treatment outcome, for each case.

## Quality assessment

We adapted the checklist developed by Hoy et al. [19] to assess the quality of records estimating prevalence or incidence. Records reporting multiple host groups (humans, intermediate or definitive hosts) were assigned a score for each host group separately. In brief, the risk of bias was assessed based on ten individual items, which were scored with 0 for 'no' (low risk of bias) and 1 for 'yes' (high risk of bias). Two members of the review team (WA and VD) assessed the quality and inconsistent results or discrepancies were solved through discussion. The total risk of bias score of each record, was defined as low (sum of scores between ≤2), moderate (sum of scores between 3 and 6), or high (sum of scores ≥7).

## Data analysis

A descriptive analysis was performed in Excel. Continuous variables were described as mean and median, while categorical variables were described as numbers and percentages. For prevalence and incidence, only records that contained relevant epidemiological information (e.g., the number of individuals examined, the number of positives and the prevalence or incidence) were considered. Due to high heterogeneity in prevalence estimates among the included studies, we did not perform a meta-analysis to estimate the national prevalence of the disease for each country in the GHA. For each prevalence estimate, 95% Wilson score confidence intervals (95%CI) were calculated using the 'binom' package in the R statistical software [20]. Finally, maps showing the geographic distribution of *E. granulosus s.l.* infections in the different hosts were generated using Quantum Geographic Information System (QGIS) software [21]. The shapefiles, constituting the foundational layer of the maps, were obtained from the openly accessible DIVA-GIS website [22].

# Results

## Literature search results and characteristics of the included records

A total of 980 records (953 from the bibliographic databases and 27 from other sources) were retrieved (Fig 1). After removal of 453 duplicate records, 527 were further screened based on their title and abstract, resulting in 374 eligible records that were subjected to full text evaluation. A total of 247 records meeting the inclusion criteria for the SR were retained (Table A in S1 File), describing 437 studies (309 in animal intermediate hosts, 79 studies in humans, and 49 studies in the definitive hosts). Of the 247 records, 177 records (representing 315 studies) were population surveys, containing estimates for the population prevalence or incidence.

The SR identified records for all countries in the GHA, except for Djibouti (Fig 2). Most of the records (212/247; 86%) described data for Ethiopia (143/247; 57.9%), Kenya (41/247; 16.6%), or Sudan (28/247; 11.3%). Similarly, most of the records (244/247; 98.8%) described data for a single country. Three records covered multiple countries: one covered Kenya and Sudan [23] and another one Kenya and Uganda [24]. In a third record, intermediate host data were collected in Saudi Arabia; however, the animals originated from Sudan and Somalia [25].

Of the 247 included records, the majority (231/247) focused on a single host group (animal intermediate hosts: 149 records; humans: 62 and definitive hosts: 20). The remaining 16 records reported data for two or three host groups. In the 437 studies described in the 247 records, cattle were the most studied species (142 studies) followed by humans (79 studies), sheep (65 studies), goats (49 studies), camel (35) and dogs (30 studies). The remaining 27 studies provide data on different wild animal definitive hosts (19), pig (9), warthog (1) and pooled data for multiple intermediate host species (8). As illustrated in S1 Fig, most records (215/247; 87.0%) were published after 2009.

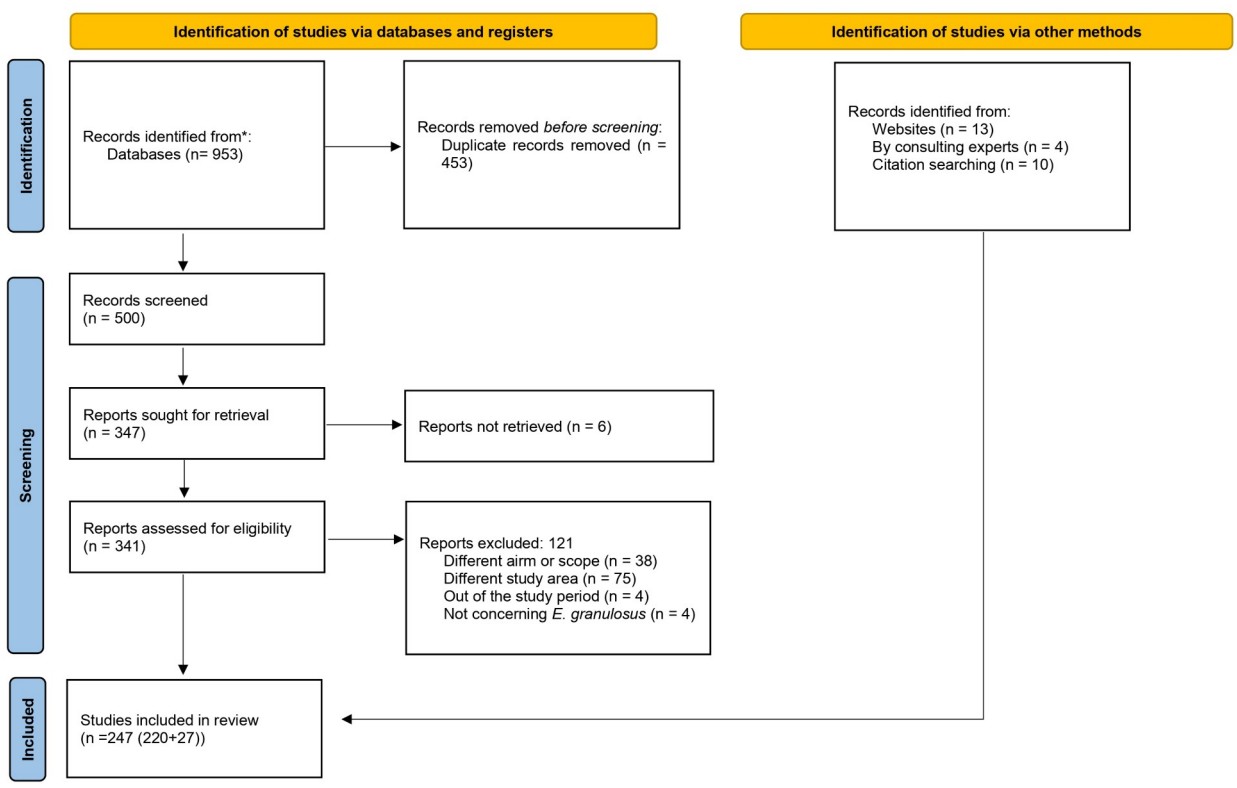

**Fig 1. PRISMA 2020 flow diagram for new systematic reviews which included searches of databases, registers and other sources.**

## Assessment of risk of bias

The quality and risk of bias were assessed for the 177 population survey records (representing 189 studies per host groups). Generally, the assessment showed that there were 25 low-quality studies (13.2%), 87 medium-quality studies (46.0%), and 77 high-quality studies (40.7%) (Table B in S1 File). For all but five of the 189 studies per host group, the target population was not representative to the national population (e.g., a specific patient group cannot be deemed representative for the general, national population). For the human population surveys, 12 out of the 16 studies (75.0%) applied a non-random sampling strategy. For the animal intermediate hosts, 137 out of the 150 studies (91.3%) were based on surveys in abattoirs, and for 66.4% (91/137) of them, the sampled population was not representative of the target population (for example, slaughtered animals were male only or originated from outside the study area). In addition, a non-random sampling strategy was applied in more than half of the studies (92/150; 61.3%). Furthermore, limitations were observed regarding the use of appropriate diagnostic tools in the studies involving intermediate hosts. In 32.0% (48/150) of the studies, only inspection of the liver and/or lungs were considered. In the studies involving the definitive host, the lack of random sampling (14/23; 60.9%) and the use of inappropriate diagnostic method (8/23; 34.8%; e.g., microscopic examination of fecal samples) were the major observed limitations.

## CE in humans

A total of 75 records, representing 79 studies, reported the occurrence of CE in humans, covering all the countries of GHA except Djibouti. These studies included population surveys (17

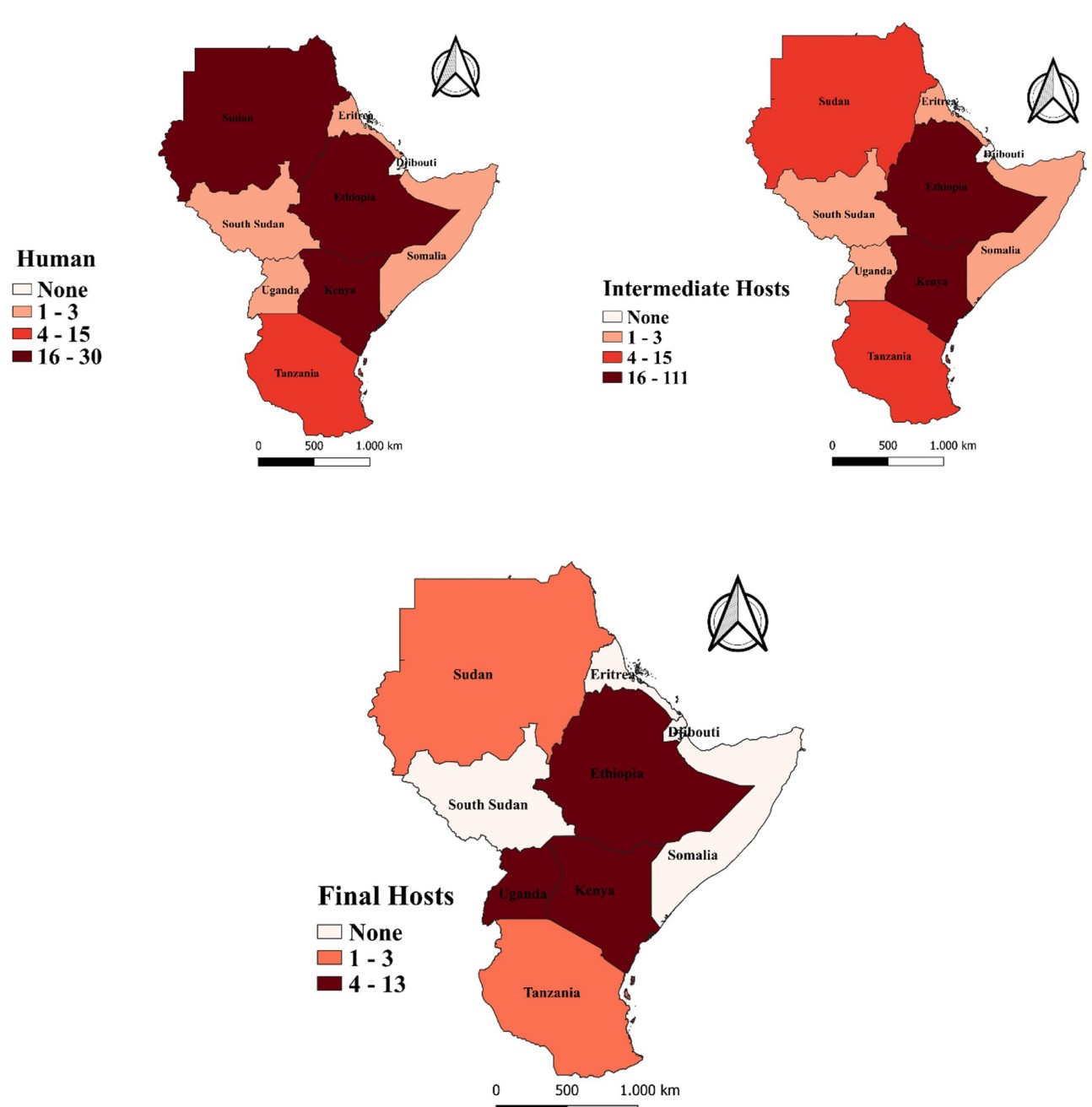

**Fig 2. Number of records on *Echinococcus granulosus s.l.* reported in the GHA between 2000 and 2022.** Shapefiles were republished from the DIVA-GIS database (https://www.diva-gis.org/gdata) under a CC BY 4.0 license, with permission from Global Administrative Areas (GADM).

studies) and case series/reports (54 studies). Another eight studies reported the molecular identification of species/genotypes, using regular polymerase chain reaction (PCR), real time PCR (rtPCR), PCR followed by restriction fragment length polymorphism (PCR-RFLP) or by PCR followed by sequencing.

**Population surveys.**   Among the 17 population survey studies, eight studies targeted the general population, whereas nine focused on hospital/clinic patients (prospective sampling:

**Table 1. Overview of human population survey studies reported in the GHA between 2000 and 2022.**

| Country | Study area | Population studied | Study period | Detection method | Sample size | No. positive | Prevalence (95% CI) | References |
|---|---|---|---|---|---|---|---|---|
| *General population* | | | | | | | | |
| Ethiopia | South Omo zone, Southern Nations, Nationalities, and Peoples' Region | Pastoral and agro-pastoral communities | 2014 | Ultrasound | 2,838 | 56 | 2.0 (1.52–2.55) | [29] |
| Kenya | Bungoma county, western Kenya | Community survey | Not available | Ultrasound | 1,002 | 7 | 0.7 (0.34–1.43) | [30] |
| Kenya | Turkana | Pastoralists/transhumant | 2010 to 2012 | Ultrasound | 7,367 | 219 | 3.0 (2.61–3.39) | [31] |
| South Sudan | Terekeka State, Mundari Tribe | Pastoralists area (cattle camps) | Not available | Ultrasound | 610 | 4 | 0.7 (0.26–1.67) | [32] |
| Sudan | Central Sudan, Gezira state (Um Zukra) | Villagers | 2002 | Ultrasound | 300 | 1 | 0.33 (0.06–1.86) | [33] |
| Sudan | Khartoum State Bahry, Omdurman and Khartoum localities | Randomly selected consenting participants | 2017 to 2018 | Serology | 305 | 20 | 6.5 (4.28–9.91) | [27] |
| Tanzania | Arusha Region | Pastoral communities | 2019 to 2020 | Ultrasound | 823 | 6 | 0.73 (0.33–1.58) | [34] |
| Uganda | Karamoja (Northeast), Teso, Central and Western regions | Pastoral and agro-pastoral communities | 2012 to 2014 | Ultrasound | 3,636 | 68 | 1.85 (1.48–2.36) | [35] |
| *Patients* | | | | | | | | |
| Ethiopia | Tikur Anbessa Hospital, Addis Ababa | Abdominal masses from patients sent for ultrasound examination | 2000 to 2001 | Ultrasound | 81 | 4 | 4.9 (1.94–12.02) | [36] |
| Sudan | Central eastern: Tampool (camel dominant area) and Nuba mountains (cattle dominant area) | Patients attending clinic | Not available | Ultrasound | 3,227 | 18 | 0.56 (0.35–0.88) | [26] |
| Tanzania | Manyara and Arusha regions in Northern Tanzania: | Inhabitants visiting randomly selected health centers | 2012 | Serology | 345 | 39 | 11.3 (8.38–15.08) | [37] |

three studies, retrospective hospital record analysis: six studies). In the general population, the prevalence based on ultrasound (US) examination ranged from 0.33 to 3.0%, whereas using serological tests, it was estimated at 6.5%. In patients visiting hospitals or local clinics, the prevalence based on serological tests was estimated at 11.3%, while the prevalence based on US ranged from 0.56 to 4.9% (Table 1). Studies estimating the incidence were restricted to Ethiopia (five studies) and Tanzania (one study) only. In Ethiopia, the annual surgical incidence rates ranged between 0.03 and 2.3 cases per 100,000 people. In Tanzania, the estimated annual surgical incidence rate equaled 10 cases per 100,000 people (Table C in S1 File).

Risk factors for human cystic echinococcosis were identified in only three studies. In Sudan, residing in a camel-dominant area, contact with dogs, and age (specifically, being older than 18 years) were reported as significant risk factors [26,27]. Additionally, a study conducted in Tanzania highlighted higher prevalence in female subjects compared to males, as well as in age groups below 30 years compared to those above 30 years [28].

**Case series and case reports.** A total of 54 studies reported either case series (n = 10) or detailed case reports (n = 44). The ten studies reporting case series were conducted in a hospital or community setting. Eight hospital-based studies reported 557 cases from Ethiopia, 736 from Kenya and 117 from Sudan, while two community-based studies reported another 527 cases in Kenya (Table D in S1 File). In the 44 detailed case reports, a total of 76 human CE cases were described, of whom 42 were males (55.4%). The CE case age ranged from 3 to 75 years with a median of 26.5 years. Most patients (56/76, 73.7%) were younger than 40 years,

**Table 2. Summary of human 76 CE cases reported in the GHA between 2000 and 2022.**

| Characteristics | No. | Proportion (in %) |
|---|---|---|
| *Sex* | | |
| Male | 42 | 55.3 |
| Female | 34 | 44.7 |
| *Age group (in years)* | | |
| 1–18 | 28 | 36.8 |
| 19–40 | 28 | 36.8 |
| 41–70 | 20 | 26.3 |
| *Patients' origin by country* | | |
| Ethiopia | 36 | 47.3 |
| Sudan | 23 | 30.3 |
| Kenya | 8 | 10.5 |
| Tanzania | 4 | 5.3 |
| South Sudan | 2 | 2.6 |
| Somalia | 2 | 2.6 |
| Eritrea | 1 | 1.3 |

including 17 paediatric cases (≤10 years). Geographically, the cases were reported throughout the GHA, except for Djibouti and Eritrea. Most of the cases were reported in Ethiopia (36/76, 47.3%) and Sudan (23/76, 30.3%) (Table 2). Of the 76 cases, eight were migrants: one Eritrean immigrant was diagnosed in Greece, while the other seven were immigrants from Ethiopia (1 case), Kenya (2 cases), Sudan (3 cases) and Somalia (1 case), were diagnosed in the USA and Australia. Out of the 16 cases for whom occupation was available, 13 of them were pastoralists or farmers, while the others were abattoir worker (1 case), housewife (1 case), or soldier (1 case) (Table E in S1 File).

In Table 3, the organ distribution, signs and symptoms, treatment methods and the outcome of human CE case reports are summarized. Although the hydatid cysts were anatomically distributed over 20 different organs of the human body, they were mainly found in lungs (25/76; 32.9%) and liver (15/76; 19.7%). Unusual anatomical sites included the breast, bladder, eye orbit, neck region, feet and ovary.

For the pulmonary CE cases, the most common clinical presentations were chest pain (15 cases), cough (15 cases) and dyspnoea (9 cases), while abdominal pain and abdominal mass were the signs and symptoms associated in case of hepatic CE (Table 3). When reported (41 out of 76 cases), the duration between onset of symptoms and diagnosis ranged from 3 days to 5 years (median: 6 months). The diagnosis was mainly based on X-ray for pulmonary CE and ultrasonography for the other organs, whereas magnetic resonance imaging (MRI) and computed tomography (CT) scans were used mainly as confirmatory diagnostic methods (S1 File **and Table E in** S1 File).

Of the 76 CE cases, 36 were treated by means of surgery only, 33 with a combination of surgery and albendazole treatment, 5 cases with an albendazole treatment only, while 2 cases were treated by means of surgery combined with both albendazole and puncture, aspiration, injection, re-aspiration (PAIR). The treatment resulted in a good prognosis except in one case (information reported for 54/76), where rupture of a cyst during the surgical intervention led to an anaphylactic shock [38].

**Species/genotype identification.** Eight studies assessed the species/genotype of 487 human CE cases from Ethiopia, Kenya, Sudan and Tanzania (Table F in S1 File). A total of 465 human cysts were molecularly investigated, of which 443 (95.3%) were identified as *E.*

**Table 3. Summary of organ distribution, clinical manifestation, treatment, and treatment outcome of 76 human CE cases reported in the GHA between 2000 and 2022.**

| Organs | No. cases | Localization (No.) | Signs and symptoms (No.) | Treatment (No.) | Outcome (No.) |
|---|---|---|---|---|---|
| Lung | 25 | Right lobes (7); left lobes (7); right hemithorax (5); bilateral lobes (3); not determined (3) | Pain (15); cough (15); dyspnoea (9); weight loss (6); fever and sweats (6); weakness (1); haemoptysis (1); upper abdominal distension (1) | Surgery (16); surgery and albendazole (8); thoracentesis and albendazole (1) | Recovery (14); recurrent dyspnoea (1); pneumothorax (1); hospital-acquired pneumonia (1); death (1); not determined (5) |
| Liver | 15 | Left lobe (3); right lobe (1); ND (11) | Pain (1); abdominal mass (1); vomiting (1); epigastric hernia (1); detected incidentally (1); not determined (11) | Surgery (8); surgery and albendazole (4); albendazole (3) | Recovery (2); postoperatively bile leak (1); not determined (12) |
| Bone | 9 | Spine (4); iliac bone (2); tibia (1); femur (1); sternum (1) | Pain (4); paraparesis and paraplegia (2); pathological fracture (2); tumor-like mass (2); loss of sensation and power (1); destructive lesion (1) | Surgery (3); surgery and albendazole (4); surgery, pair and albendazole (2) | Recovery (9) |
| Brain | 5 | cerebrum (5) | Headache (4); hemiparesis (4); nausea and vomiting (4); weakness (1); visual impairment and seizure disorder (1) | Surgery/craniotomy (4); surgery and albendazole (1) | Recovery (4); recurrence due to rupture of the primary cyst during surgery (1) |
| Heart | 5 | Interventricular septum (3); right pleural cavity & interventricular septum (1); right heart/right ventricle (1) | Exertional angina (1); haemoptysis and shortness of breath (1); syncopal episode (1); chronic cough (1); heart murmur (1), body weight loss (1); right cardiac failure (1); palpitation and decreased exercise tolerance (1) | Surgery and albendazole (4); surgery/ cystopericystectomy followed by thoracotomy (1) | Recovery (3); not determined (2) |
| Spleen | 2 | posteriorly facing the diaphragm (1); not determined (1) | Pain (1); abdominal swelling (1); left hypochondrial dull aching pain (1); left hypochondrial distension (1) | Surgery (splenectomy) and albendazole (1); surgery (splenectomy) and mebendazole (1) | Recovery (1); abscess with wound infection and wound dehiscence (1) |
| Muscle | 2 | Right thigh (1); left thigh (1); right lower back musculoskeletal | Cystic swelling and pain (1); asymptomatic (detected incidentally during CE survey) (1) | Surgery (1); surgery and albendazole (1) | Recovery (2) |
| Pancreas | 1 | In the body and tail of the pancreas | Epigastric mass | Surgery (xipho-umbilical incision) | Postoperative pneumonia |
| Kidneys | 1 | Right kidney | Tenderness in the right hypochondrium and lumbar region | Surgery and albendazole | Recovery |
| Neck | 1 | Left lateral lower neck | Swelling | Surgery/excisional biopsy and albendazole | Recovery |
| Eye | 1 | Right orbit | Painless, progressive proptosis of the right eye | Surgical excision | Recovery |
| Bladder | 1 | Left side | Inability to pass urine and fever | Surgery and albendazole | Recovery |
| Ovary | 1 | Right ovary | Lower abdominal swelling and irregular menstruation | Surgery (exploratory laparotomy) | Recovery |
| Breast | 1 | Left breast | Painless swelling | Surgery and albendazole | Recovery |
| Feet | 1 | Plantar surface | Swelling associated with pain | Surgery and albendazole | Recovery |
| Heart and Lung | 1 | Myocardium of RV and right lung lower lobe | Cough; breathing difficulties; fever; emaciation | Fluid drainage using chest tube and albendazole | Patient was lost to follow-up. |
| Abdominal and pelvic cavities | 1 | Right side abdominal and pelvic cavities (1) | Shortness of breath and fatigue; abdominal mass | Albendazole | Patient was lost to follow-up. |
| Lung and Kidneys | 1 | Bilateral of both lungs and Kidneys (1) | Cough and chest pain (1) | Albendazole and supportive therapy | Recovery |
| Bone and Liver | 1 | Spine (1) | Spastic paraparesis (1) | Surgery and albendazole | Recurrence |

*granulosus s.s.* (G1, G3, $G_{Omo}$) and 21 (4.5%) as *E. canadensis* (G6/7). $G_{Omo}$, a novel *E. granulosus s.s.* genotype, was reported from the southern part of Ethiopia. The described genotype was phylogenetically positioned in the *E. granulosus s.s.*/*E. felidis* clade. It is closely related to *E. granulosus s.s.*, but clearly distinct from G1 and G3 [2]. In Kenya, both *E. granulosus s.s.* and *E. canadensis* (G6/7) were detected, with *E. granulosus s.s.* being the most common (440/445; 98.9%) [24,39,40]. In Tanzania, two *E. granulosus s.s.* species were reported [34] while in Sudan, *E. canadensis* (G6/7) was the only species detected in 14 cases [41–43]. No other species (e.g. *E. ortleppi*) were reported for the study area.

## Animal intermediate hosts

A total of 165 records, representing 309 studies, investigated the presence of *E. granulosus s.l.* in animal intermediate host species, covering all the study countries except Djibouti. The majority of the the studies (249) were population surveys, while the remaining 60 studies reported identification of species/genotypes (59 studies) using molecular tests including both PCR and rtPCR using different primers, restriction fragment length polymorphism PCR (RFLP-PCR), and sequencing. Additionally, one study presented a case report.

**Population surveys.** Of the 249 population survey studies, 246 were abattoir-based and the remaining three were based on ultrasound (two) and serology (one). Among the 246 studies involving abattoir sampling, 180 performed the examination of all suspected organs using the standard inspection methods for CE (detailed carcass inspection), 34 studies were based on abattoir record analysis (routine carcass inspection) and 32 studies were based on the examination of lungs and/or liver only (Table 4). Most of the studies (124/249, 49.8%) were performed on cattle from Ethiopia, Kenya, Sudan, Uganda, and Tanzania, followed by sheep (50/249, 20.1%) and goats (41/249, 16.5%) from all the study countries, except Djibouti. Twenty-two and seven studies were done on camels and pigs, respectively, in Ethiopia, Kenya, and Sudan (Table 4). In general, the reported prevalence estimates ranged between 0 and 61.6% in camels, 0.01 to 88.4% in cattle, 0 to 65.2% in goats, 0.05 to 10% in pigs, and 0 to 67.8% in sheep (Table 4). Another five studies reported one pooled prevalence estimate for multiple species (Table G in S1 File).

Putative risk factors for CE in the animal intermediate hosts were reported by 78 studies. Among these, age (36 studies), body condition score (BCS) (19 studies), sex (11 studies), breed (5 studies), origin of the animals (3 studies), production system (2 studies) and management (2 studies) were the reported risk factors (Table H in S1 File). Studies assessing associations between age and CE reported a higher prevalence in older animals. In studies considering BCS as a risk factor, prevalence was higher in animals with a poor or medium BCS compared to animals with a good BCS. In most of the studies reporting sex as a risk factor (8/11; 73%), female animals were at higher risk of infection. Furthermore, a higher prevalence was reported in local breeds compared to exotic or cross breeds in most studies reporting breed as a risk factor (4/5, 80%).

Regarding the anatomical distribution of CE in the organs, lungs and liver were the most commonly affected organs in all host species (Table I in S1 File). Lungs were the organs most affected in camels (57.2%), cattle (57.0%), and pigs (52.4%), while in sheep cysts were mainly found in the liver (61.4%). In goats, both lung (46.8%) and liver (46.9%) were equally affected. Other organs affected were spleen, heart, kidney, carcass, tongue, and muscles.

The fertility of cysts was tested in a total of 104 studies in the different host species (Table J in S1 File). The fertility rate (proportion of the cysts being fertile) ranged from 21.0 and 80.0% (with a mean of 25.7%). The viability of the fertile cysts was investigated in a total of 55 studies. Overall, 51.8% of cysts were found to be viable, however the viability ranged between 48.1 for cattle and 70.4% for goats (Table 5).

**Table 4. Prevalence ranges of CE in animal intermediate hosts species reported in the GHA between 2000 and 2022.** [a]Exported animals from Sudan and Somalia to Saudi Arabia revealing zero prevalence estimates.

| Host (No. studies) | Country (No. studies) | Diagnostic method | | | | | | | |
|---|---|---|---|---|---|---|---|---|---|
| | | Detailed carcass inspection | | Routine carcass examination | | Lung and/or Liver inspection | | Other | |
| | | No. studies | Prevalence range (%) | No. studies | Prevalence range (%) | No. studies | Prevalence range (%) | No. studies | Prevalence range (%) |
| ***Camels (22)*** | | | | | | | | | |
| | Ethiopia (11) | 9 | **6–61.6** | – | – | **2** | **21.2–30.8** | – | – |
| | Kenya (4) | 3 | **6.9–61.4** | – | – | **1** | **25.3** | – | – |
| | Sudan (7) | 4 | **0 ᵃ– 59.9** | – | – | **3** | **4.6–20** | – | – |
| ***Cattle (124)*** | | | | | | | | | |
| | Ethiopia (91) | 73 | **5.1–88.4** | **10** | **7.5–56.7** | 7 | **5–52.6** | 1 (serology) | **43.3** |
| | Kenya (14) | 8 | **0.9–25.8** | **3** | **0.9–50.4** | 3 | **4.2–6.0** | – | – |
| | South Sudan (1) | 1 | **3.9** | – | – | – | – | – | – |
| | Sudan (5) | 4 | **2.5–6.1** | – | – | 1 | **0.01** | – | – |
| | Tanzania (12) | 6 | **1.6–48.7** | **4** | **3.1–42.3** | 2 | **2.9–3.2** | – | – |
| | Uganda (**1**) | – | – | – | – | 1 | **4.9** | – | – |
| ***Goats (41)*** | | | | | | | | | |
| | Ethiopia (19) | 15 | **0–65.2** | **2** | **11.0–28.0** | 2 | **1.4–4.8** | – | – |
| | Kenya (11) | 7 | **0.4–15.2** | **2** | **2.0–34.3** | 1 | **1.6** | 1 (ultrasound) | **1.8** |
| | South Sudan (2) | 2 | **2.7–24.9** | | | | | | |
| | Sudan (2) | 1 | **1.9** | – | – | – | – | 1 (ultrasound) | **4.3** |
| | Tanzania (6) | 2 | **22.2–34.7** | **2** | **5.9–48.1** | 2 | **1.5–4.1** | – | – |
| | Uganda (**1**) | 1 | **33.3** | – | – | – | – | – | – |
| ***Pigs (7)*** | | | | | | | | | |
| | Ethiopia (1) | 1 | **9.9** | – | – | – | – | – | – |
| | Kenya (4) | 1 | **5.0** | **2** | **0.7–2.4** | 1 | **0.05** | | |
| | Tanzania (2) | 2 | **0.4–4.3** | – | – | – | – | – | – |
| ***Sheep (50)*** | | | | | | | | | |
| | Ethiopia (25) | 21 | **1.1–67.8** | **2** | **5.4–8** | 2 | **4–8.5** | – | – |
| | Kenya (11) | 8 | **1.5–16.5** | **2** | **0.1–48.8** | 1 | **1.3–4.5** | – | – |
| | Somalia (1) | 1 | **0 ᵃ** | – | – | – | – | – | – |
| | South Sudan (1) | 1 | **7.0** | – | – | – | – | – | – |
| | Sudan (4) | 4 | **0.6–11.3** | – | – | – | – | – | – |
| | Tanzania (7) | 3 | **16.6–63.8** | **2** | **5.1–9.6** | 2 | **1.5–3.5** | – | – |
| | Uganda (1) | 1 | **42.5** | – | – | – | – | – | – |

**Case reports.** There was one case report of CE in an animal intermediate host (camel) in Sudan. The lungs of the 10 years old female camel contained 43 cysts of various sizes, ranging from 3 to 17 cm in diameter [44].

**Species/genotype identification.** The species/genotype was assessed in 59 studies describing the molecular investigation results for a total of 3,369 hydatid cysts. Four species were identified, namely *E. granulosus s.s.* (G1/3), *E. ortleppi* (G5), *E. canadensis* (G6/7) and *E. felidis*. Among these, the two major zoonotic species *E. granulosus s.s.* (G1/3) and *E. canadensis* (G6/7) represented 54.8% and 42.9% of hydatid cysts investigated, respectively. Both species were

**Table 5. The fertility and viability of *Echinococcus s.l.* cysts in animal intermediate hosts reported in the GHA between 2000 and 2022.**

| Host species | Fertility test | | | | Viability test | | | |
|---|---|---|---|---|---|---|---|---|
| | N studies | N cysts examined | N fertile cyst | Percent of fertile cyst | N studies | N fertile cyst examined | N viable | Percent of viable cyst |
| Camel | 12 | 4,711 | 2,758 | 58.5 | 6 | 859 | 576 | 67.0 |
| Cattle | 58 | 38,551 | 8,085 | 21.0 | 34 | 6,283 | 3,023 | 48.1 |
| Goat | 12 | 821 | 284 | 34.6 | 5 | 125 | 88 | 70.4 |
| Pig | 1 | 5 | 4 | 80.0 | 1 | 4 | 2 | 50.0 |
| Sheep | 21 | 4,716 | 1,406 | 29.8 | 9 | 639 | 422 | 66.0 |
| **Overall** | **104** | **48,828** | **12,537** | **25.7** | **55** | **7,910** | **4,111** | **51.9** |

reported in Ethiopia, Kenya, Sudan, Tanzania, and Uganda. *Echinococcus ortleppi* was detected in Ethiopia, Kenya, Sudan, and Tanzania, though at a low prevalence (2.3% of investigated hydatid cysts). *Echinococcus felidis* was detected in only one study in Uganda (Table 6).

In camels, *E. canadensis* (G6/7) was the dominant species detected in Kenya and Sudan. In cattle, sheep, and goat *E. granulosus s.s.* (G1/3) was dominant in Ethiopia and Kenya, while in Sudan, *E. canadensis* (G6/7) was the dominant species. The detection of *Echinococcus felidis* and *E. granulosus s.s.* (G1/3) was reported in a warthog (*Phacochoerus* spp.) in Uganda (Table K in S1 File).

## Definitive hosts

A total of 25 records, describing 49 studies, investigated the presence of *E. granulosus s.l.* in the definitive host species (dogs and wildlife) from Ethiopia, Kenya, Sudan, Tanzania and Uganda. All of the 49 studies were population surveys.

**Population surveys.** Of the 49 studies, 30 investigated the prevalence of *E. granulosus s.l.* in dogs. These studies were conducted in Ethiopia, Kenya, Sudan, Tanzania and Uganda. The most surveyed dog populations were stray dogs (12 studies), followed by free roaming owned (9 studies) and confined dogs (8 studies). Other dog populations were dogs presenting at veterinary clinics (2 studies), and dog sampled through environmental fecal samples (2 studies) (Table L in S1 File). The prevalence varied for the different diagnostic techniques applied: microscopy (3.6 to 94.5%), necroscopic examination (16.7 to 88.9%), serology (12.4 to 26.0%) and molecular detection (0.6 and 47.6%) using PCR, PCR-RFLP and sequencing (Table 7). However, the molecular prevalence among the microscopic or necroscopic positive taeniid eggs or worms ranged from 10.5 to 93% (S1 File **and Table L in** S1 File).

Nineteen studies investigated the presence of *E. granulosus s.l.* in various wildlife species, in Ethiopia, Kenya Tanzania and Uganda. The prevalence varied according to the diagnostic techniques used: being microscopy (4.3 to 69.2%), serological (0 to 37.5%) and molecular (0 to 100%) methods. Moreover, the prevalence varied widely among the different wildlife species and countries (Table 7).

Only one study from Kenya reported risk factors associated with infection in definitive hosts: age of the animals was found to be a risk factor, younger dogs being at higher risk for infection [45].

**Species/genotype identification.** Nine studies reported the identification of *E. granulosus s.l.* species/genotypes for a total of 198 definitive host samples (Table 8). In dogs, species/genotype identification was reported only in Kenya and Sudan. In Kenya, five species, namely *E. granulosus s.s, E. ortleppi, E. canadensis* (G6/7), *E. felidis* and *E. equinus* were detected. *E. granulosus s.s.* and *E. canadensis* (G6/7), were the dominant species. In one study from Sudan, *E.*

**Table 6. Species/genotypes distribution of CE in animal intermediate host species reported in the GHA between 2000 and 2022.**

| Host (No. studies) | Country (No. studies) | CE species/genotypes | | | | |
|---|---|---|---|---|---|---|
| | | *E. granulosus s.s.* (G1/3); N (%) | *E. ortleppi* (G5); N (%) | *E. canadensis* (G6/7); N (%) | *E. felidis*; N (%) | Total |
| *Camel (12)* | | | | | | |
| | Ethiopia (3) | 36 (70.6) | – | 15 (29.4) | – | 51 |
| | Kenya (4) | 55 (15.7) | – | 296 (84.3) | – | 351 |
| | Sudan (5) | – | 1 (0.2) | 422 (99.8) | – | 423 |
| *Cattle (18)* | | | | | | |
| | Ethiopia (5) | 272 (94.4) | 11 (3.8) | 5 (1.7) | – | 288 |
| | Kenya (6) | 511 (81.6) | 37 (5.9) | 78 (12.5) | – | 626 |
| | Sudan (5) | 2 (0.8) | 10 (3.9) | 246 (95.3) | – | 258 |
| | Tanzania (2) | 57 (86.4) | 8 (12.1) | 1 (1.5) | – | 66 |
| *Goats (8)* | | | | | | |
| | Ethiopia (1) | 3 (37.5) | – | 5 (62.5) | – | 8 |
| | Kenya (6) | 194 (71.6) | 5 (1.8) | 72 (26.6) | – | 271 |
| | Sudan (1) | – | – | 67 (100) | – | 67 |
| *Sheep (15)* | | | | | | |
| | Ethiopia (3) | 16 (100) | – | – | – | 16 |
| | Kenya (7) | 611(97.1) | 3 (0.5) | 15 (2.4) | – | 629 |
| | Sudan (4) | – | – | 205 (100) | – | 205 |
| | Tanzania (1) | 7 (100) | | | | 7 |
| *Pig (2)* | | | | | | |
| | Ethiopia (1) | 1 (50) | 1 (50) | – | – | 2 |
| | Kenya (1) | 2 (50) | 1 (25) | 1 (25) | – | 4 |
| *Warthog (1)* | | | | | | |
| | Uganda (1) | 1 (50) | – | – | 1 (50) | 2 |
| *Sheep and Goat (1)* | | | | | | |
| | Ethiopia (1) | 4 (100) | – | – | – | 4 |
| *Sheep, Goat and Camel (1)* | | | | | | |
| | Sudan (1) | – | – | 15 (100) | – | 15 |
| *Sheep, Goat, Cattle and Camel (1)* | | | | | | |
| | Uganda (1) | 73 (96.1) | – | 3 (3.9) | – | 76 |
| **Overall** | | 1,845 (54.8) | 77 (2.3) | 1,446 (42.9) | 1 (0.003) | 3,369 |

*canadensis* (G6/7) was identified in all dogs but one, in which *E. granulosus s.s.* was found (Table M in S1 File). In Uganda, one study reported the detection of *E. granulosus s.l.* without reporting the specific species/genotype [46].

In wild animals, three studies reported species/genotype identification in Kenya and Uganda. In Kenya both *E. granulosus s.s.* and *E. felidis* were reported, while only *E. felidis* was reported in Uganda (Table 8 and Table M in S1 File). In Ethiopia, one study confirmed the molecular detection of *E. granulosus s.l.* in an Ethiopian wolf (*Canis simensis*), however, the specific species/genotype was not identified [47].

## Discussion

As part of the sustainable development goal 3 (good health and well-being), WHO commits to end the epidemics of various infectious diseases, including the neglected zoonotic tropical diseases such as CE [7]. To accomplish these WHO targets, in-depth knowledge on the

**Table 7. The prevalence of *E. granulosus s.l.* in definitive host species reported in the GHA between 2000 and 2022.**

| Host (No. studies) | Country (No. studies) | Diagnostic method | | | | | | | |
|---|---|---|---|---|---|---|---|---|---|
| | | Microscopy | | Necropsy | | Serology | | Molecular | |
| | | No. studies | Prevalence range | No. studies | Prevalence range | No. studies | Prevalence range | No. studies | Prevalence range |
| **Dog (30)** | | | | | | | | | |
| | Ethiopia (14) | 8 | **3.6–94.5** | 6 | **16.7–90.0** | – | – | – | – |
| | Kenya (8) | 3 | **5.5–11** | 1 | 33 | 1 | 26 | 3 | **0.6–18.0** |
| | Sudan (3) | 1 | **44** | 1 | 51.2 | – | – | 1 | 47.6 |
| | Tanzania (2) | 1 | **6.4** | – | – | 1 | 12.4 | – | – |
| | Uganda (3) | 1 | **15.0** | 1 | 66.3 | – | – | 1 | 12.2 |
| **Banded mongoose (*Mungos mungo*) (1)** | | | | | | | | | |
| | Tanzania (1) | – | – | – | – | 1 | 0 | – | – |
| **Bat eared fox (*Otocyon megalotis*) (1)** | | | | | | | | | |
| | Tanzania (1) | – | – | – | – | 1 | 14.3 | – | – |
| **Black backed jackal (*Canis mesomelas*) (1)** | | | | | | | | | |
| | Tanzania (1) | – | – | – | – | 1 | 0 | – | – |
| **Cheetah (*Acinonyx jubatus*); (1)** | | | | | | | | | |
| | Tanzania (1) | – | – | – | – | 1 | 16.1 | – | – |
| **Ethiopian Wolf (*Canis simensis*) (3)** | | | | | | | | | |
| | Ethiopia (3) | 2 | **4.3–12.5** | – | – | – | – | 1 | 100 |
| **Genet (*Genetta tigrina*); (1)** | | | | | | | | | |
| | Tanzania (1) | – | – | – | – | 1 | 0 | – | – |
| **Leopard (*Panthera pardus*) (2)** | | | | | | | | | |
| | Kenya (1) | – | – | – | – | – | – | 1 | 14.3 |
| | Uganda (1) | | | | | | | 1 | 0 |
| **Lion (*Panthera leo*) (5)** | | | | | | | | | |
| | Kenya (1) | – | – | – | – | – | – | 1 | 47.5 |
| | Tanzania (1) | – | – | – | – | 1 | 10 | – | – |
| | Uganda (3) | 1 | **69.2** | – | – | – | – | 2 | **72–100** |
| **Serval cat (*Leptailurus serval*) (1)** | | | | | | | | | |
| | Tanzania (1) | – | – | – | – | 1 [a] | 0 | – | – |
| **Spotted hyaena (*Crocuta crocuta*) (3)** | | | | | | | | | |
| | Kenya (1) | – | – | – | – | – | – | 1 | 30.0 |
| | Tanzania (1) | – | – | – | – | 1 | 37.5 | – | – |
| | Uganda (1) | – | – | – | – | – | – | 1 | 20.0 |

epidemiology of these diseases in endemic areas is primordial to develop effective control interventions. GHA is considered to be endemic for *E. granulosus s.l.*, but a detailed overview of the current epidemiology across the different hosts was lacking. Here, we systematically reviewed recent literature on the occurrence and risk factors of *E. granulosus s.l.* in both humans and animals in the GHA.

## The GHA is confirmed to be a textbook example for *E. granulosus s.l.* transmission

Based on the population surveys retained in our SR, the reported prevalence estimates ranged up to 11.3% in humans, 61.6% in camels, 88.4% in cattle; 65.2% in goats, 9.9% in pigs, 67.8% in sheep and 94.5% in dogs. Although less surveyed (19 out of the 49 studies that investigated *E.*

**Table 8. Species/genotypes distribution of *E. granulosus s.l.* in definitive host species reported in the GHA between 2000 and 2022.**

| Host (No. studies) | Country (No. studies) | *E. granulosus s.l.*; N (%) | | | | | |
|---|---|---|---|---|---|---|---|
| | | *E. granulosus s.s.* (G1/3) | *E. ortleppi* (G5) | *E. canadensis* (G6/7) | *E. felidis* | *E. equinus* (G4) | Total positive |
| **Dog (3)** | | | | | | | |
| | Kenya (2) | 51 (61.4) | 5 (6.0) | 23 (27.7) | 3 (3.6) | 1 (1.2) | 83 |
| | Sudan (1) | 1 (2.5) | – | 39 (97.5) | – | – | 40 |
| **Leopard (1)** | | | | | | | |
| | Kenya (1) | 1 (100) | – | – | – | – | 1 |
| **Lion (3)** | | | | | | | |
| | Kenya (1) | 8 (27.6) | – | – | 21 (72.4) | – | 29 |
| | Uganda (2) | – | – | – | 35 (100) | – | 35 |
| **Spotted hyena (2)** | | | | | | | |
| | Kenya (1) | 3 (33.3) | – | – | 6 (66.7) | – | 9 |
| | Uganda (1) | – | – | – | 1 (100) | – | 1 |
| **Overall (9)** | | 64 (32.3) | 5 (2.5) | 62 (31.3) | 66 (33.3) | 1 (0.5) | 198 |

*granulosus s.l.* in the definitive host species), *E. granulosus s.l.* was also found in various wild canids and felines that are endemic to the GHA. For humans, these reported prevalence estimates are higher than the global prevalence range estimate [5], as well as those described for Europe [48] and southern Africa [49], comparable with northern Africa [10], and lower than those reported for the Middle East [50]. We also confirm the country-level occurrence of *E. granulosus s.l.* for all host groups as reported by both the WHO [7] and the World Organization for Animal Health (WOAH) [51]. An exception occurred for animal cases for Djibouti, Sudan and South Sudan. Compared to the reports of WOAH indicating the presence of *E. granulosus s.l.* in domestic and wild animals (without specifying whether they refer to intermediate or definitive hosts) in Djibouti, our SR either did not retrieve any records for animal intermediate hosts or definitive hosts from the country. On the other hand, our SR provided evidence of occurrence of *E. granulosus s.l.* in animal intermediate and definitive hosts for both Sudan and South Sudan, whereas the WOAH reports it as absent. Our enhanced efforts to screen a comprehensive of literature resources including grey literature as well as the fact that the WOAH reports are based on reports of national Veterinary Services as opposed to scientific studies, could explain the expanded evidence. For Djibouti, it remains unclear why we missed the evidence, but it could be due to lack of sufficient veterinary and public health research in the country.

## An accurate assessment of the epidemiology based on the existing recent literature remains challenging

Despite the confirmation of the GHA being an endemic zone for *E. granulosus s.l.*, our SR has also demonstrated that an accurate assessment of the epidemiology based on the existing recent literature remains challenging, due to the lack of well-designed studies. To start, the risk of bias assessment indicated that less than half of population survey studies (40.7%) were of high quality, with the most commonly observed issues being the surveyed population not being representative for the national population (97.4%) or not being randomly selected (62.4%). As a consequence, reported estimates might be biased. For example, some studies focused on sampling pastoral communities, e.g., in Ethiopia [29], Kenya [31] and Tanzania [34], and the reported prevalence estimates may thus be an overestimation for the true underlying national prevalence. Other examples are studies reporting annual surgical incidence

rates. Given that people will not always seek medical attention, the true underlying incidence is probably underestimated. In addition to this, literature is severely imbalanced towards specific host species and countries. Across the different host species, cattle were the most surveyed (32.5%; 142 out of 437 studies). Similarly, more than half of the records reported findings in one country, namely Ethiopia (57.9%). Finally, a wide range of diagnostic tools have been applied across the different studies, each with their own, yet imperfect clinical sensitivity and specificity. An extreme example is the use of microscopic examination of stool of the definitive hosts, which does not allow for the differentiation of *E. granulosus s.l.* eggs from other taeniids [52].

## A firm response against *E. granulosus s.l.* in the GHA will require more ambitious, interdisciplinary and transboundary partnerships

Although by the zoonotic nature of CE it is obvious that interdisciplinary investigations applying the One Health principles are essential, for the studies retrieved in our SR, only two records [24,53] assessed the occurrence of *E. granulosus s.l.* across all the actors of the life cycle (humans, intermediate and definitive host species) in the same geographical areas. This gap was also highlighted in a recent scoping review on zoonoses research in the Horn of Africa [11]. Moreover, none of the SR studies performed an in-depth investigation of risk factors, including of behavioural aspects and animal management practices as was done for other areas (e.g., dog ecology analysis in Kyrgyzstan, [54], therefore the necessary scientific basis to develop targeted intervention strategies remains lacking. Moreover, given that the life cycle involves wildlife endemic to the GHA and that *E. granulosus s.l.* is highly endemic among pastoral communities, it will be equally important to engage in transboundary partnerships. The need for this is already apparent from the studies reporting on hyperendemic foci in adjacent areas of Ethiopia, Kenya, South Sudan, and Tanzania that are inhabited by traditional pastoralists of various ethnic groups. Examples for these hyperendemic foci are Turkana County, which is in the northwest of Kenya, bordering with South Sudan and Ethiopia, and Maasai land, which ranges from north of Tanzania to the south of Kenya [28,31,34,38]. Other examples for the need of transboundary partnerships, are the mobility of animals from endemic to non-endemic areas which may result with possible translocation of species/genotypes [55].

Overall, to inform both public health and animal health decision makers, both at the national (e.g., ministries of health and agriculture) and the international level (e.g., WHO and WHAO), more studies with appropriate and harmonized designs (e.g., diagnosis), and more in-depth interdisciplinary and transboundary partnerships, an in depth-assessment of the true health and socio-economic impact of CE in the GHA, will be needed. This is evident from the fact that, although all countries in the GHA apart from Somalia, have performed a zoonosis or transboundary diseases prioritization processes [56], only one, being Ethiopia, identified echinococcosis as a zoonosis to be prioritized [57]. However, echinococcosis was the second most studied zoonosis in a recent scoping review in the Horn of Africa [11]. The health impact of CE should be quantified by means of Disability Adjusted Life Years; an established WHO public health metric widely applied to express burden of disease [58]. The socio-economic impact assessment should include direct as well as indirect costs related to human (e.g., costs for treatment; costs due to absence from work) and animal CE (e.g., costs for treatment and economic losses due to condemnation of offal). Such an exercise would assist in optimizing the allocation of resources towards the fight against this important disease.

Despite the strengths of our study in terms of conducting a SR comprising all the involved hosts in the GHA, and our efforts to retrieve a comprehensive literature as well, our SR also had some limitations. First, our language restriction might have caused us missing relevant records

written in other languages than English and French. Moreover, due to a large variation in prevalence estimates between and within diagnostic tests, study designs, study quality and period of study we did not perform a meta-analysis to estimate the national and region-wide prevalence of the disease. Finally, as inherent to any SR, the estimated prevalence ranges rely heavily on the study quality of the included studies, and therefore they might be misleading for the true prevalence ranges of CE in the GHA, due to the use of imperfect tests by the included studies, focus on certain hyperendemic foci and limitations in the study design of the included studies.

## Supporting information

**S1 Checklist. PRISMA 2020 checklist.**
(DOCX)

**S2 Checklist. A template designed for consulting experts for grey literature.**
(DOCX)

**S1 Fig. Distribution of included records in the systematic review of *E. granulosus* infection in the GHA by publication year (2000 to 2022).**
(XLSX)

**S1 File.  (Tables A to M): Table A.** Studies included in the SR reporting the occurrence of *E. granulosus* infection in the GHA. **Table B.** Risk of bias score for each study per host group. **Table C.** Surgical incidence of human CE based on hospital records. **Table D.** Hospital case series and community survey studies of human CE without reporting prevalence or incidence rates. **Table E.** Human CE case reports data. **Table F.** Studies reporting species/genotype identification of human CE from cross-sectional and case report studies. **Table G.** Population survey studies of CE in each species of animal intermediate host. **Table H.** Population survey studies that reported risk factors for the occurrence of CE in an animal intermediate host. **Table I.** Anatomical distribution of CE in the organs of infected animal intermediate host species. **Table J.** Cyst fertility and viability among the different animal intermediate hosts. **Table K**. Studies reporting species/genotype identification of *E. granulosus* in animal intermediate host species. **Table L.** Population survey studies of *E. granulosus* in the definitive hosts species. **Table M.** Studies reporting species/genotype identification of *E. granulosus* in the definitive host species.
(XLSX)

## Acknowledgments

We wish to thank the Institute of Tropical Medicine (ITM) library staff members for their help in retrieving Full text portable document format (PDF) records that were not freely accessible online.

## Author Contributions

**Conceptualization:** Weldegebrial G. Aregawi, Bruno Levecke, Hagos Ashenafi, Nigatu Kebede, Pierre Dorny, Veronique Dermauw.

**Data curation:** Weldegebrial G. Aregawi, Bruno Levecke, Veronique Dermauw.

**Formal analysis:** Weldegebrial G. Aregawi, Veronique Dermauw.

**Funding acquisition:** Weldegebrial G. Aregawi, Bruno Levecke, Pierre Dorny, Veronique Dermauw.

**Methodology:** Weldegebrial G. Aregawi, Bruno Levecke, Veronique Dermauw.

**Project administration:** Bruno Levecke.

**Resources:** Charles Byaruhanga, Erastus Mulinge, Marion Wassermann, Thomas Romig.

**Supervision:** Bruno Levecke, Pierre Dorny, Veronique Dermauw.

**Writing – original draft:** Weldegebrial G. Aregawi, Bruno Levecke, Hagos Ashenafi, Charles Byaruhanga, Nigatu Kebede, Erastus Mulinge, Marion Wassermann, Thomas Romig, Pierre Dorny, Veronique Dermauw.

**Writing – review & editing:** Weldegebrial G. Aregawi, Bruno Levecke, Hagos Ashenafi, Charles Byaruhanga, Nigatu Kebede, Erastus Mulinge, Marion Wassermann, Thomas Romig, Pierre Dorny, Veronique Dermauw.

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
