## [Decision Letter · Decision Letter 0]

9 Aug 2023

Dear Dr Aregaw,

Thank you very much for submitting your manuscript "Epidemiology of Echinococcus granulosus sensu lato in the Greater Horn of Africa: A systematic review" for consideration at PLOS Neglected Tropical Diseases. As with all papers reviewed by the journal, your manuscript was reviewed by members of the editorial board and by several independent reviewers. In light of the reviews (below this email), we would like to invite the resubmission of a significantly-revised version that takes into account the reviewers' comments. 

Your manuscript has been reviewed by two independent reviewers. They recommended major revisions to your manuscript and I concur. Please see the points carefully and address them and resubmit your manuscript. Thanks for your cooperation.

Reviewer #1: This paper is well written realizing a systematic review of the data about E. granulosus sensu lato in a specific area of Africa.

The introduction gave a clear overview of the state of the art for CE in GHA. The protocol of the systematic review is relevant, well described and properly applied. The results are interesting and discussed in a relevant way, being aware of the limitations of the study due to the data obtained.

I have only some minor comments for this very interesting paper.

l. 90: close the brackets after Uganda

l. 94: please also close the brackets

Regarding species/genotype identification for both humans and animals, it may be interesting to specify the different methods used especially to confirm (or not) the high specificity.

l. 293-303: no case of E. ortleppi in humans? As it is known as zoonotic, it may be interest to briefly discuss this absence of report even some cases probably exist.

l. 331-332: poor or medium BCS is observed infected animals, but is it considered as a consequence of CE or a risk factor to develop infection?

l. 337-339: as the organ most frequently infected is not the same regarding the intermediate host species, is it possible to combine with molecular identification of the CE species to confirm or not the existing data?

The same question regarding fertility among studies, which can provide both data (species and fertility of cysts). The aim as for organs infected is to check if there are no differences with global observations.

l. 371: I suggest to rather used definitive than final for the title

l. 404: it is five species described not five genotypes

l. 432-434: it would be interesting to compare the prevalence in humans in GHA with those observed in North-Africa which is known to be highly endemic with numerous data available.

l. 455: please suppress the second "s" to obtain focused

Maybe useful to indicate that a standardization or harmonization of the diagnostic tests would be also useful.

Reviewer #2: A systematic review of the literature on Echinococcus granulosus sensu lato species in Africa, as well as the rest of the world, is essential to advance knowledge, inform policy and implement effective control and prevention measures for echinococcosis, ultimately leading to improved global public health outcomes.

In the present study, a total of 980 records were retrieved from a literature search, of which 247 met the inclusion criteria for the systematic review. The majority of the records described data from Ethiopia, Kenya and Sudan. Quality and risk of bias were assessed for 177 population-based studies, with 40.7% of the studies being of high quality. The prevalence of CE in humans ranged from 0.33% to 3.0% based on ultrasonography and 6.5% based on serology. In animals, the prevalence ranged from 0.01% to 88.4% in cattle, 0 to 65.2% in goats, 0.05% to 10% in pigs and 0 to 67.8% in sheep. E. granulosus s.s. was the most common genotype found in humans and animals. The lack of well-designed studies and the wide range of diagnostic tools used make it difficult to accurately assess the epidemiology of CE in the Greater Horn of Africa.

Overall, the study was conducted and the data analysis and interpretation were performed in a good, transparent and understandable manner. The manuscript is well written and understandable. However, there are some points that should be at least partially improved regarding the epidemiological and statistical data analysis. See below for details:

1. What is generally missing is meta-analysis. When prevalence data from several studies are combined, this is the most common and powerful method. Meta-analysis combines estimates from individual studies to produce an overall summary estimate of prevalence, often using statistical techniques such as random-effects or fixed-effects models. Meta-analysis can provide a more precise and robust estimate of prevalence and can assess heterogeneity between studies. 

a. Forest plots would be useful to display the results of individual studies together with the pooled summary estimate from the meta-analysis. Forest plots allow you to visualise the variability between studies and to see the overall trend. 

b. Subgroup analysis, which is more detailed from a statistical point of view, would also provide clearer and possibly deeper insights if there are substantial differences between studies. 

c. Meta-regression could also be done, because you have specific covariates (e.g. age, sex, study quality) that may affect the prevalence estimates, you can do a meta-regression to explore their potential effects.

d. A cumulative meta-analysis might provide information on whether there is a trend in the prevalence data over time...

2. Another suggestion would be to calculate the 95% confidence intervals and add them to the calculated prevalences (e.g. lines 338-339, Table 2). 

3. Lines 330-331: Here it is not clear how significance was tested. This missing information should be added with appropriate p-values.

Kind regards

We cannot make any decision about publication until we have seen the revised manuscript and your response to the reviewers' comments. Your revised manuscript is also likely to be sent to reviewers for further evaluation.

Sincerely,

jong-Yil Chai

Academic Editor

Eva Clark

Section Editor

Your manuscript has been reviewed by two independent reviewers. They recommended major revisions to your manuscript and I concur. Please see the points carefully and address them and resubmit your manuscript. Thanks for your cooperation.

Reviewer #1: This paper is well written realizing a systematic review of the data about E. granulosus sensu lato in a specific area of Africa.

The introduction gave a clear overview of the state of the art for CE in GHA. The protocol of the systematic review is relevant, well described and properly applied. The results are interesting and discussed in a relevant way, being aware of the limitations of the study due to the data obtained.

I have only some minor comments for this very interesting paper.

l. 90: close the brackets after Uganda

l. 94: please also close the brackets

Regarding species/genotype identification for both humans and animals, it may be interesting to specify the different methods used especially to confirm (or not) the high specificity.

l. 293-303: no case of E. ortleppi in humans? As it is known as zoonotic, it may be interest to briefly discuss this absence of report even some cases probably exist.

l. 331-332: poor or medium BCS is observed infected animals, but is it considered as a consequence of CE or a risk factor to develop infection?

l. 337-339: as the organ most frequently infected is not the same regarding the intermediate host species, is it possible to combine with molecular identification of the CE species to confirm or not the existing data?

The same question regarding fertility among studies, which can provide both data (species and fertility of cysts). The aim as for organs infected is to check if there are no differences with global observations.

l. 371: I suggest to rather used definitive than final for the title

l. 404: it is five species described not five genotypes

l. 432-434: it would be interesting to compare the prevalence in humans in GHA with those observed in North-Africa which is known to be highly endemic with numerous data available.

l. 455: please suppress the second "s" to obtain focused

Maybe useful to indicate that a standardization or harmonization of the diagnostic tests would be also useful.

Reviewer #2: A systematic review of the literature on Echinococcus granulosus sensu lato species in Africa, as well as the rest of the world, is essential to advance knowledge, inform policy and implement effective control and prevention measures for echinococcosis, ultimately leading to improved global public health outcomes.

In the present study, a total of 980 records were retrieved from a literature search, of which 247 met the inclusion criteria for the systematic review. The majority of the records described data from Ethiopia, Kenya and Sudan. Quality and risk of bias were assessed for 177 population-based studies, with 40.7% of the studies being of high quality. The prevalence of CE in humans ranged from 0.33% to 3.0% based on ultrasonography and 6.5% based on serology. In animals, the prevalence ranged from 0.01% to 88.4% in cattle, 0 to 65.2% in goats, 0.05% to 10% in pigs and 0 to 67.8% in sheep. E. granulosus s.s. was the most common genotype found in humans and animals. The lack of well-designed studies and the wide range of diagnostic tools used make it difficult to accurately assess the epidemiology of CE in the Greater Horn of Africa.

Overall, the study was conducted and the data analysis and interpretation were performed in a good, transparent and understandable manner. The manuscript is well written and understandable. However, there are some points that should be at least partially improved regarding the epidemiological and statistical data analysis. See below for details:

1. What is generally missing is meta-analysis. When prevalence data from several studies are combined, this is the most common and powerful method. Meta-analysis combines estimates from individual studies to produce an overall summary estimate of prevalence, often using statistical techniques such as random-effects or fixed-effects models. Meta-analysis can provide a more precise and robust estimate of prevalence and can assess heterogeneity between studies. 

a. Forest plots would be useful to display the results of individual studies together with the pooled summary estimate from the meta-analysis. Forest plots allow you to visualise the variability between studies and to see the overall trend. 

b. Subgroup analysis, which is more detailed from a statistical point of view, would also provide clearer and possibly deeper insights if there are substantial differences between studies. 

c. Meta-regression could also be done, because you have specific covariates (e.g. age, sex, study quality) that may affect the prevalence estimates, you can do a meta-regression to explore their potential effects.

d. A cumulative meta-analysis might provide information on whether there is a trend in the prevalence data over time...

2. Another suggestion would be to calculate the 95% confidence intervals and add them to the calculated prevalences (e.g. lines 338-339, Table 2). 

3. Lines 330-331: Here it is not clear how significance was tested. This missing information should be added with appropriate p-values.

Kind regards
---

## [Decision Letter · Decision Letter 1]

2 Jan 2024

Dear Dr Aregaw,

We are pleased to inform you that your manuscript 'Epidemiology of Echinococcus granulosus sensu lato in the Greater Horn of Africa: A systematic review' has been provisionally accepted for publication in PLOS Neglected Tropical Diseases.

Best regards,

jong-Yil Chai

Academic Editor

Eva Clark

Section Editor

Your revised manuscript has been reviewed by 2 independent reviewers. Both of them agreed to accept your manuscript and I concur Thanks for your kind cooperation.

Reviewer's Responses to Questions

**Key Review Criteria Required for Acceptance?**

**Methods**

-Are the objectives of the study clearly articulated with a clear testable hypothesis stated?

-Is the study design appropriate to address the stated objectives?

-Is the population clearly described and appropriate for the hypothesis being tested?

-Is the sample size sufficient to ensure adequate power to address the hypothesis being tested?

-Were correct statistical analysis used to support conclusions?

-Are there concerns about ethical or regulatory requirements being met?

Reviewer #1: As previously reported the method is relevant and correctly applied accordint to the type of data available

Reviewer #3: I agree with the reviewers of the initial version of the manuscript that methodology used is correct

**Results**

-Does the analysis presented match the analysis plan?

-Are the results clearly and completely presented?

-Are the figures (Tables, Images) of sufficient quality for clarity?

Reviewer #1: The results are clearlly presented

Reviewer #3: I agree with the reviewers of the initial version of the manuscript in most comments. It is disappointing that authors did not take in consideration the statistical analysis suggested by reviewer #2, however, the explanation given by the authors seems sufficient given the quality of the data.

**Conclusions**

-Are the conclusions supported by the data presented?

-Are the limitations of analysis clearly described?

-Do the authors discuss how these data can be helpful to advance our understanding of the topic under study?

-Is public health relevance addressed?

Reviewer #1: The conclusions are in accordance ith the limitations of the study known by the authors as reported in the paper but also in a relevant way in their answers to reviewers comments.

Reviewer #3: The conclusions are supported by the data, the authors also explain all disadvantages of the review which is appreciated.

**Editorial and Data Presentation Modifications?**

Reviewer #1: none

Reviewer #3: brackets should not be italicized with scientific names.

The readers not familiarized with Africa would appreciate a brief explanation of the name Greater Horn of Africa

Please briefly explain why the authors only review data from January 2000 to October 2022.

In Table #3 why using the word cerebrum and not brain?

**Summary and General Comments**

Reviewer #1: All the comments were integrated or a relevant answer was provided justifying to not realize modifications.

Reviewer #3: Overall, the review is well written and concise, it provides important information of the epidemiology of CE in several African countries.

PLOS authors have the option to publish the peer review history of their article (what does this mean?). If published, this will include your full peer review and any attached files.

Reviewer #1: No

Reviewer #3: No

---

## [Editor Report · Acceptance letter]

10 Jan 2024

Dear Dr Aregawi,

We are delighted to inform you that your manuscript, "Epidemiology of Echinococcus granulosus sensu lato in the Greater Horn of Africa: A systematic review," has been formally accepted for publication in PLOS Neglected Tropical Diseases.

Best regards,

Shaden Kamhawi

co-Editor-in-Chief

Paul Brindley

co-Editor-in-Chief
